# Systematic comparison of sea urchin and sea star developmental gene regulatory networks explains how novelty is incorporated in early development

Gregory A. Cary [1,3,5], Brenna S. McCauley[1,4,5], Olga Zueva[1], Joseph Pattinato[1], William Longabaugh[2] & Veronica F. Hinman [1✉]

The extensive array of morphological diversity among animal taxa represents the product of millions of years of evolution. Morphology is the output of development, therefore phenotypic evolution arises from changes to the topology of the gene regulatory networks (GRNs) that control the highly coordinated process of embryogenesis. A particular challenge in understanding the origins of animal diversity lies in determining how GRNs incorporate novelty while preserving the overall stability of the network, and hence, embryonic viability. Here we assemble a comprehensive GRN for endomesoderm specification in the sea star from zygote through gastrulation that corresponds to the GRN for sea urchin development of equivalent territories and stages. Comparison of the GRNs identifies how novelty is incorporated in early development. We show how the GRN is resilient to the introduction of a transcription factor, *pmar1*, the inclusion of which leads to a switch between two stable modes of Delta-Notch signaling. Signaling pathways can function in multiple modes and we propose that GRN changes that lead to switches between modes may be a common evolutionary mechanism for changes in embryogenesis. Our data additionally proposes a model in which evolutionarily conserved network motifs, or kernels, may function throughout development to stabilize these signaling transitions.

[1] Department of Biological Sciences, Carnegie Mellon University, Pittsburgh, PA 15213, USA. [2] Institute for Systems Biology, Seattle, WA, USA. [3]Present address: The Jackson Laboratory, Bar Harbor, ME, USA. [4]Present address: Huffington Center on Aging, Baylor College of Medicine, 1 Baylor Plaza, Houston, TX 77030, USA. [5]These authors contributed equally: Gregory A. Cary, Brenna S. McCauley. ✉email: vhinman@andrew.cmu.edu

The regulatory program that controls development is unidirectional and hierarchical. It initiates with early asymmetries that activate highly coordinated cascades of gene regulatory interactions known as a gene regulatory network (GRN). GRNs function to orchestrate the intricate cellular and morphogenic events that comprise embryogenesis[1,2], and their topologies must be structured in ways that permit the robust development needed to reliably produce viable embryos. While genetic variation can arise anywhere in the genome and affect any part of an individual GRN, the need to form a viable embryo provides a constraint on the types of variation that pass the filter of selection. The timing and mechanisms of potential developmental constraint persist as topics of intense debate[3–7], which can only be resolved by systems-level comparisons of experimentally established GRNs. The evolution of transcription factors (TFs) used in early development presents an especially intriguing problem in the context of maintaining developmental stability[8].

The GRN for the specification of sea urchin endomesoderm is the most comprehensive, experimentally derived GRN known to date[9–11]. It explains how vegetal-most micromeres express signaling molecules, including Delta, needed to specify the adjacent macromere cells to endomesoderm, how micromeres ingress as mesenchyme, and are finally specified to form a biomineralized skeleton. This GRN initiates with the maternally directed nuclearization of β-catenin, which activates the paired homeodomain TF *pmar1*[12]. Pmar1 represses the expression of *hesC*. The HesC TF is a repressor of genes encoding many of the TFs needed to specify micromere fate (i.e., *alx1*, *ets1*, *tbr*, and *tel*) including the *delta* gene. The activation of Pmar1, therefore, indirectly leads to the expression of many of the regulatory genes within the vegetal pole, micromere territory in what has been termed the double-negative gate[13]. The Pmar1 TF appears to be a novel duplication of the *phb* gene, and is found only in sea urchins[14]. No clear ortholog of *pmar1* exists in available genomes or transcriptomes from sea stars (*Patiria miniata* and *Acanthaster planci*), brittle stars (*Amphiura filiformis*), or hemichordates (*Saccoglossus kowalevskii*)[15–19] and thus the Pmar1 repression of *hesc*, i.e., the double-negative gate[13], functions only in modern sea urchins.

In this work, we assemble a detailed GRN for sea star endomesoderm specification through gastrulation (Supplementary Fig. 1). Understanding the impact of integrating novelty into early development demands such a systems-level approach: not one limited to local properties around the new circuit, but an understanding of how the network as a whole responds to the change. An interactive, temporal model, including primary and published data, is hosted on a web server (grns.biotapestry.org/PmEndomes), which allows for further and more fine-grained exploration (Supplementary Fig. 2). This GRN was produced using the same experimental approaches as those used to generate the sea urchin network[20] to allow for a meaningful comparison. This sea star GRN, therefore, presents an unprecedented opportunity to compare these networks to understand how they have evolved. Using this network comparison, we show how bimodal switches in signaling pathways permit an evolutionary transition in regulatory network topology, and how such a transition is buffered by the presence of conserved regulatory kernels.

## Results

### Delta and HesC are engaged in a canonical regulatory motif. 
In contrast to the sea urchin[13] (Fig. 1f), sea star blastula stage embryos co-express the *delta* and *hesC* transcripts throughout the endomesoderm-fated vegetal pole (Fig. 1a), but their expression is partitioned into adjacent cells by midway through gastrulation (Fig. 1b). Indeed, in all known species of echinoderms that lack an identifiable *pmar1* gene, *hesC* remains expressed within the *delta*+

territory of the blastula mesoderm[15,17,21]. The related Phb genes have recently been shown to act as positive inputs into endomesodermally expressed *hesc* in sea star and cidaroid sea urchin[14]. We show that Tgif also positively regulates *hesC* expression specifically in the sea star mesoderm (Fig. 1e). This input is not possible in the sea urchin given that these transcription factors are not co-expressed[22,23]. Not all inputs into *hesC* are changed, however, as *blimp1* (formerly *krox*), a known repressor of *hesC* in sea urchin[24], also represses sea star *hesC* (Fig. 1c, d). Thus, the gain of repression by Pmar1 and the loss of positive input from Tgif drive the differences in *hesC* and *delta* co-expression between sea urchins and sea stars (Fig. 1g). The local impact of integrating *pmar1* in the sea urchin network, therefore, is the exclusion of *hesC* expression from the territory of cells that will express the *delta* ligand at the vegetal pole. It is via the Delta signal that these cells induce adjacent cells to various endomesodermal fates[25–28]. The early asymmetry in expression of the Delta ligand makes sea urchin micromeres sufficient to induce ectopic endomesoderm when transplanted to the animal pole or to animal blastomeres alone[29]. Thus, Delta induction is critical to specifying mesodermal cell types and is one of the central genes providing the regulatory function of the micromeres in sea urchins.

The co-expression of transcripts encoding Delta, HesC, and the Notch receptor (Supplementary Fig. 3) in the sea star vegetal pole suggests the potential for lateral inhibition regulatory interactions. In many contexts lateral inhibition functions to segregate an equipotential field of cells into distinct cell types[30–33]. Perturbing Delta-Notch signaling and the expression of *hesC* in sea star blastulae reveals that signaling between adjacent cells through the Notch receptor activates the expression of *hesC* (Fig. 2a–c), which in turn represses *delta* expression (Fig. 2d, e). Thus, we demonstrate that the change in upstream regulation between sea urchin and sea stars that results in co-expression of *delta* and *hesC* at blastula stage allows for lateral inhibition (LI) regulatory interactions in sea stars, compared to the inductive mechanism used in sea urchins (Fig. 2h).

### Partitioning of mesodermal subtypes involves Delta/Notch lateral inhibition. 
Given that Delta, Notch, and HesC engage in canonical LI, we sought to understand how this process might function in specifying cell types in the sea star mesoderm. The sea star mesoderm originates from the central vegetal pole of the late blastula, a molecularly uniform territory (Fig. 3a, b and Supplementary Fig. 4). During gastrulation this territory sits at the top of the archenteron and segregates into at least two distinct cell types —blastocoelar mesenchyme and coelomic epithelium (Supplementary Fig. 4). These two lineages become molecularly distinct by mid-gastrula stage (~36 h) and *ets1* expressing mesenchyme cells begin ingressing at 46 h (Fig. 3c–f). Each *ets1*+ cell is generally separated from another *ets1*+ cell by two intervening nuclei of *ets1*− cells (Fig. 3g) while the intervening cells express *six3* (Fig. 3h), a gene that is also initially broadly mesodermal in blastulae, but is later expressed in the coelomic epithelium (Supplementary Fig. 3). *ets1*+ cells also express the transcript encoding the Delta ligand (Fig. 3i). From these data we propose a model in which lateral inhibition leads to the restricted expression of *ets1* in the *delta*+ cell and *six3* in the neighboring cell.

We inhibited Notch signaling to explicitly test the lateral inhibition model. The model predicts that such a treatment would lead to an expansion of the primary cell type (i.e., *delta*+ presumptive mesenchyme) and a concomitant reduction of the secondary cell type (i.e., presumptive coelomic mesoderm). Indeed, inhibition of Notch results in an increase in cells expressing the mesenchyme genes *ets1* and *erg* (Fig. 4a–f), and a

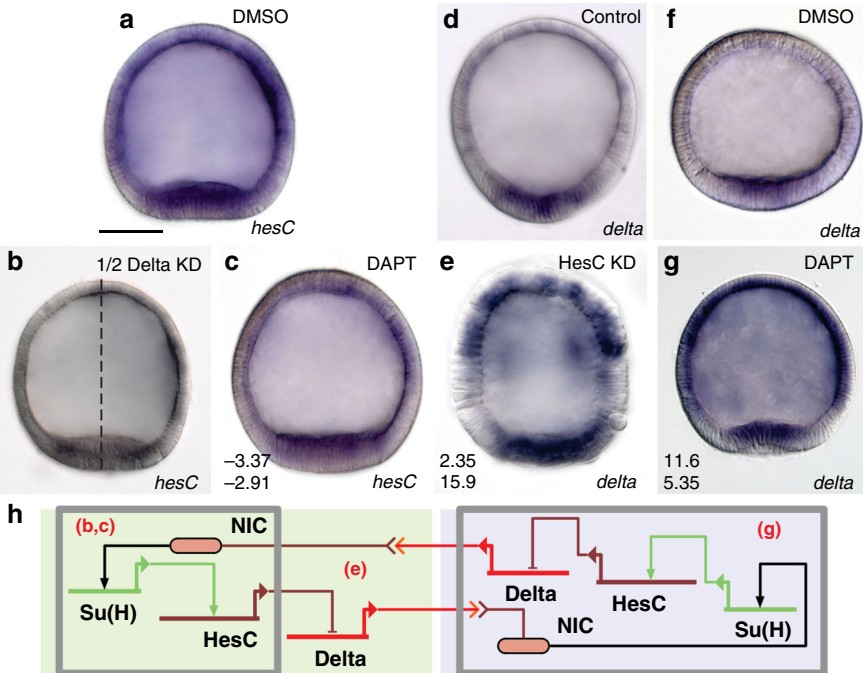

**Fig. 1 Sea star *hesC* is positively regulated downstream of the mesoderm kernel and is co-expressed with *delta*.** Sea star *hesC* and *delta* transcripts are co-expressed in the vegetal mesoderm (**a**) until mid-gastrula stage (**b**), or 36 hours post fertilization (hpf). Sea star *hesC* and *blimp1* are expressed in partially non-overlapping domains during blastula stage (**c**) and morpholino knockdown (KD) of *blimp1* results in an expansion of the expression domain of *hesC* (**d**). Morpholino knockdown of sea star Tgif produces a mesoderm-specific decrease in *hesC* expression (**e**). Schematic showing non-overlapping expression domains of sea urchin *hesC* and *delta* transcripts at blastula stage (**f**). Regulatory inputs to the *hesC* gene that are specific to sea urchin embryos, sea star embryos, and those that are common to both (**g**). Data shown are double fluorescent WMISH showing both *hesC* expression (green) and either expression of *delta* or *blimp1* (magenta) (**a**–**c**) and colorimetric WMISH (**d**, **e**). Data are representative of two biologically independent experiments consisting of at least ten embryos each. Scale bar represents 50 μm; applicable to all images in the panel.

**Fig. 2 Testing lateral inhibition of *delta* and *hesC* by inhibition of Notch signaling (DAPT) and morpholino knockdown (KD) of HesC.** Using DAPT, an inhibitor of the proteolytic gamma-secretase necessary for notch signal transduction, we observe both a down-regulation of *hesC* (**c**) and an upregulation of *delta* transcripts (**g**). Importantly the down-regulation of *hesC* is phenocopied by injection of a morpholino targeting the *delta* transcript into one of the first two blastomeres (**b**). Knockdown of HesC with an antisense morpholino yields an upregulation of both *delta* (**e**) and *hesC* transcripts. Lateral inhibition network showing relationships tested by previous experiments (**h**); red letters indicate figure panel above supporting connection. All images are colorimetric WMISH with the probes to the indicated genes. Images are representative of two biologically independent experiments consisting of at least ten embryos each. Scale bar represents 50 μm; applicable to all images in the panel. Numbers in the lower left corner of (**c**, **e**, and **g**) represent normalized log2 fold-change values of perturbed expression compared with control (i.e., **a**, **d**, and **f**) as measured by qPCR.

reduction in cells expressing coelomic epithelium genes *six3* and *pax6* (Fig. 4g–j). Moreover, we also observe a consistent morphological shift with an increase in the number of cells ingressing into the blastocoel from the archenteron and a reduction in the epithelium. These data confirm our hypothesis that the sea star mesoderm partitions through the action of Delta-Notch-mediated lateral inhibition. This also shows a consequence of the changes to the regulation of *hesC* that allow co-expression with *delta*—a switch in the mode of Delta-Notch signaling between lateral inhibition when co-expressed and induction when spatially distinct. Asymmetric expression of Delta-Notch regulators typically produces an inductive mode of signaling[34], and these results suggest that the incorporation of Pmar1 into the early sea urchin network has contributed to this switch.

**Conserved Six3-Pax6 circuit is necessary for appropriate coelomogenesis.** In sea star embryos, Delta-Notch LI segregates mesenchyme from celom. While *hesC* expression is associated with cells fated to the celom, *hesC* expression is no longer detected in the mesoderm by 48 h (Supplementary Fig. 5e), shortly after the completion of cell type partitioning and coincident with the onset of epithelial-mesenchymal transition. The Delta-Notch LI must then "hand-off" to another set of genes to stabilize and maintain coelomic restricted gene expression. Correspondingly, this stage is also the first time that we observe the expression of *pax6* and *eya* in this territory[35]. *pax6*, *eya*, *six1/2*, *dach*, and *six3* are expressed in coelomic-fated mesoderm in both sea stars and sea urchins along with other genes from the highly conserved retinal determination gene

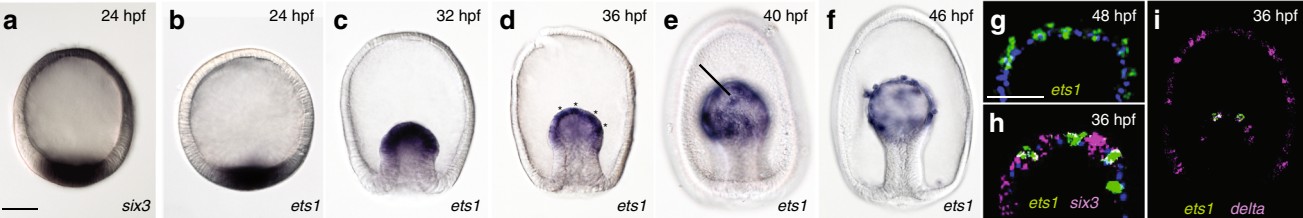

**Fig. 3 Segregation of mesodermal subtypes into interleaved cells by 36 hpf.** Sea star *ets1* and *six3* transcripts are co-expressed in the vegetal mesoderm of blastula stage embryos (**a**, **b**) at 24 h post fertilization (hpf). The expression of the *ets1* transcript was assessed every 2 h from the onset of gastrulation by colorimetric WMISH. At 32 hpf the expression of *ets1* is uniform throughout the mesoderm (**c**). At 36 hpf there is a discontinuity in the expression of *ets1* transcript (**d**, asterisks). Patches of *ets1* expression become more distinct by 40–42 hpf (**e**, line), and *ets1* expressing cells start to ingress beginning at 46 hpf (**f**). Cells expressing *ets1* transcript (green) at the tip of the archenteron are adjacent to cells with no detectable *ets1* expression (**g**), using fluorescent WMISH with a DAPI counterstain (blue), and are interleaved by cells expressing *six3* transcript (magenta) (**h**). Cells expressing *ets1* transcript also express the transcript encoding the Delta ligand (**i**). Data are representative of two biologically independent experiments consisting of at least ten embryos each. Scale bars represent 50 μm; scale bar in (**a**) applicable to (**a**–**f**) and (**i**), scale bar in (**g**) applicable in (**g**–**h**), which show a magnified region at the tip of the archenteron.

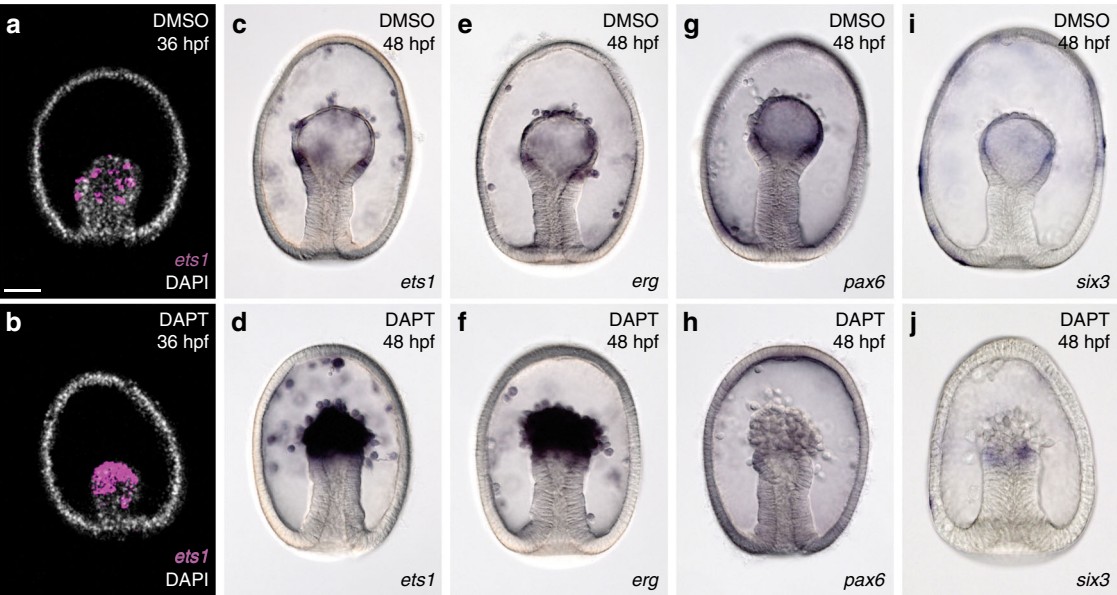

**Fig. 4 Testing the lateral inhibition model of mesodermal subtype segregation.** The expression of *ets1* transcript (magenta) appears in a salt-and-pepper distribution throughout the mesoderm at 36 h post fertilization (hpf) by fluorescent WMISH (**a**) with DAPI stained nuclei (white). Treatment with the Notch inhibitor DAPT beginning at the 2-cell stage results in a uniform expression of *ets1* in this territory (**b**). By 48 hpf, the mesenchyme cells expressing *ets1* and *erg* (**c**, **e**) ingress into the blastocoel while cells that do not ingress express *pax6* and *six3* (**g**, **i**). DAPT treatment results in an increase in cells expressing *ets1* and *erg* (**d**, **f**) and a reduction in cells expressing *pax6* and *six3* (**h**, **j**). There is also a consistent morphological shift with an increase in the number of ingressing cells and a reduction in the epithelium when Notch signaling is blocked. Data shown in (**c**–**j**) are colorimetric WMISH. Data are representative of two biologically independent experiments consisting of at least ten embryos each. Scale bar represents 50 μm; applicable to all images in the panel.

network (RDGN)[35,36]. In the sea urchin celom, these genes are wired together in a network topology considered to be homologous to that of the RDGN[37]. We examined the regulatory interactions between *pax6*, *six3*, *eya*, *dach*, and *six1/2* to determine if a similar network is involved in the maintenance of sea star coelomic mesoderm. We find a similar activation of *six3* by Pax6, of *six1/2* by itself, and of *eya* by Pax6, Six3, and Dach (Fig. 5a). Thus, these genes interact in a similar regulatory subnetwork in both sea star and sea urchin coelomic mesoderm, and this subcircuit is highly similar to the *Drosophila* RDGN (Fig. 5b), suggesting even deeper conservation of this network architecture. While Delta-Notch-mediated lateral inhibition is responsible for early segregation of mesenchymal and coelomic cell fates in sea stars, a Pax6 and Six3-mediated network is necessary for proper coelomogenesis after the completion of lateral inhibition (Supplementary Fig. 5).

## Discussion

We present a comprehensive GRN for the specification of sea star endomesoderm; the new data presented here links the previously described GRN for the early specification of endoderm and mesoderm in the sea star to the later events during gastrulation. This GRN was produced using the same experimental approaches as those used to generate the sea urchin network[20] to allow for a meaningful comparison; i.e., whole-mount in situ hybridization (WMISH) to determine spatiotemporal gene expression, and quantitative reverse transcription PCR (qRT-PCR) and WMISH in control and morpholino antisense oligonucleotide and small molecule inhibitor-induced knockdown of protein function. This network, comprising 42 nodes and 84 edges, approaches in scope the GRN for endomesoderm specification of equivalent stages in sea urchin[9], at present comprising 72 nodes and 271 edges. Although gene perturbation by morpholino knockdown has

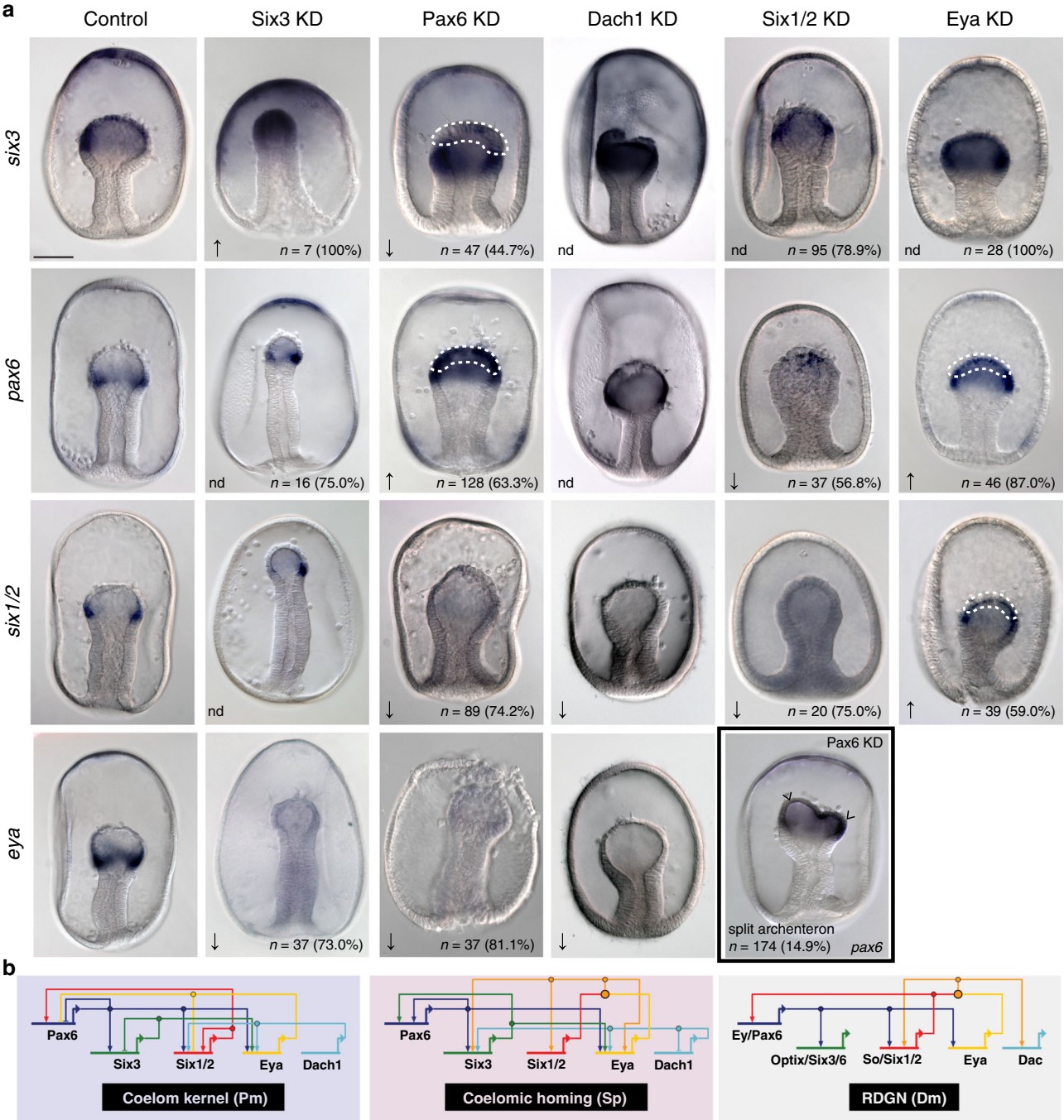

**Fig. 5 Subcircuit including *pax6*, *six3*, *eya*, *dach1*, and *six1/2* involved in sea star coelomogenesis. a** Data shown are colorimetric WMISH using the probes indicated on the left in the conditions listed along the top. Scale bar represents 50 μm; applicable to all images in the panel. *six3* expression is normally distributed throughout the mesodermal bulb of the archenteron at 48 hpf, while *pax6*, *six1/2*, *dach1*, and *eya* are normally expressed at the posterior aspect of the mesodermal bulb, having been cleared earlier from anterior regions of the mesoderm. Phenotypic effect of the perturbation of each gene is indicated, including no difference (nd), increase (↑), decrease (↓). The number of embryos assessed and percent of embryos expressing the phenotype are also reported. Some reported phenotypes are localized to the top of the archenteron and are highlighted (dashed line); e.g., the effect of Pax6 knockdown on *six3* expression is reported specifically for the anterior region of archenteron (dashed line). Some Pax6 knockdown embryos exhibited a bifurcated archenteron (e.g., boxed panel, "split archenteron"). Data are representative of two biologically independent experiments consisting of at least ten embryos each and specific phenotype counts are detailed in Supplementary Table 3. These results enabled the construction of a sub-network for sea star coelomic epithelium (**b**) and we note a similar regulatory sub-network in both sea star and sea urchin coelomic mesoderm, which is strikingly similar to the retinal determination gene network (RDGN) in *Drosophila*.

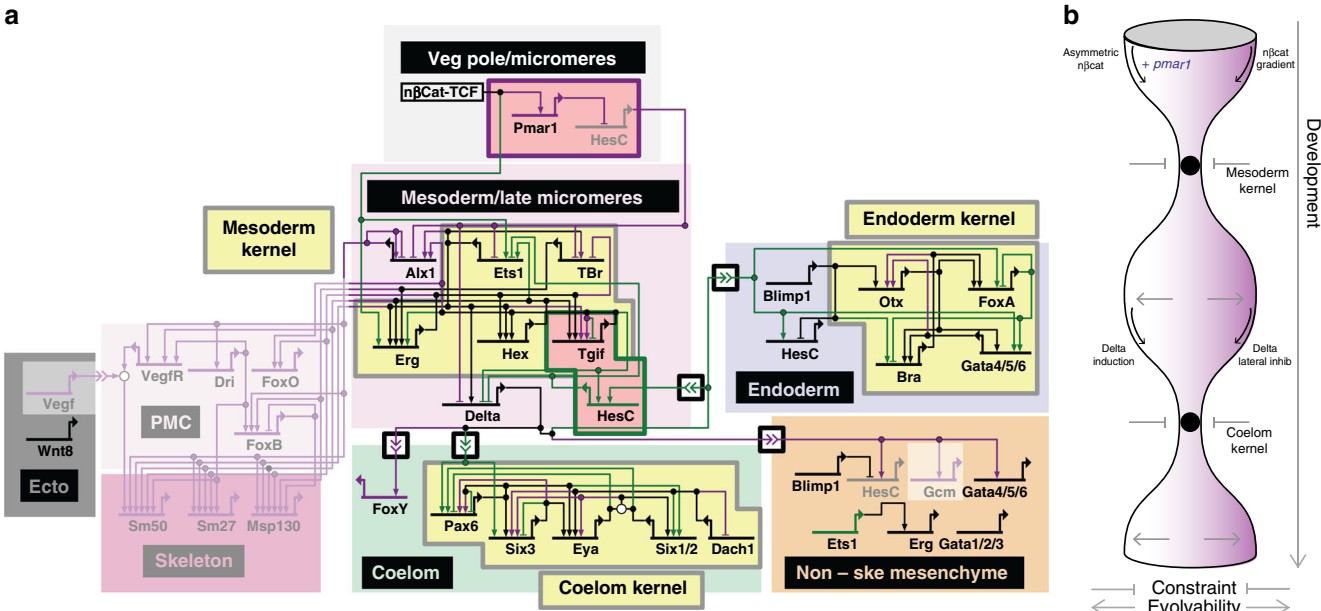

**Fig. 6 Evolutionary constraint of network kernels permits alterations to the surrounding network.** A synthesis of key aspects of the GRN from sea urchins and sea stars is shown (**a**). Genes (nodes) are shown in the territories (colored boxes) in which they are expressed. Edges show regulation by the originating upstream factor and are either positive (arrow) or repressive (bar). Signaling across cell types are indicated as double arrow heads, and Delta to Notch signals are boxed. Genes and links that are unique to sea urchin embryos are colored purple, those specific to sea stars are green, and those in common are black. Network kernels are highlighted (yellow) as are distinct sub-circuits (pink), including the sea urchin-specific double-negative gate (i.e., Pmar and HesC, purple outline) and sea star-specific positive regulation of HesC (i.e., Tgif and HesC, green outline). Grayed out backgrounds indicate entire network circuits that are absent in sea stars. Our model of GRN evolution is depicted (**b**) showing that network kernels are constrained regions whereas both up and down the hierarchy the network is capable of change.

historically been a useful tool for disentangling these networks, it is important to recognize that the links drawn are largely provisional until fully borne out by independent perturbation methods (e.g. CRISPR) and cis-regulatory analyses to test whether the indicated interactions are indeed direct.

The summary view of these results permits a global comparison of these echinoderm GRN topologies (Fig. 6A), which are the synthesis of over a decade of work including the present study. Immediately apparent are the several distinct subcircuits found in common between these networks. Common modules include *ets1, erg, hex*, and *tgif* in the mesoderm[38,39]; *otx, gatae, bra*, and *foxa* in the endoderm[40,41]; and *pax6, eya, six3, dach*, and *six1/2* in the celom (Fig. 5[37]). In contrast, entire subcircuits present in sea urchins, i.e., *dri, foxb*, and *vefgr* that direct batteries of skeletogenic differentiation genes in the sea urchin micromeres, are entirely absent from the sea star network. This comparison also reveals that similarly regulated subcircuits are highly positively cross regulated, in keeping with the previous definition of network kernel[42]. It was previously suggested that such kernels would be found in early development, as they function downstream of maternally derived and transient signals, to stabilize gene expression needed to specify distinct embryonic territories[43]. A stabilizing function is thought to be derived from the intracircuit positive regulatory feedback. Here we show that these kernels appear throughout the GRN and are not limited to only early development. Preliminary experimental analyses from other species of echinoderms[15,44–46] suggest these kernels are present in multiple species and thus represent a genuinely conserved, rather than convergent, feature of GRNs. The mechanistic basis for the evolutionary stability of these subcircuits remains unclear and it will be important to define additional such network motifs to begin to understand whether the observed stability is a cause or a consequence of the observed highly recursive regulatory wiring of these motifs[42].

From these data we propose a model of how changes in the GRN are incorporated while maintaining an overall network stability. We have detailed how the network incorporates novel circuitry into early development, in this example, the Pmar1-HesC double-negative gate. These networks use the same signaling pathways at the same places in the GRN but utilize different signaling modalities of the pathways; the networks use binary versus dosage dependence of nβ-catenin[47], and Delta induction versus Delta-Notch lateral inhibition. We argue that changes in GRNs, such as the introduction of novel genes or subcircuits, that lead to switching between alternate, stable modes of signaling pathways may be a common source of evolutionary change in these GRNs. The disruption caused by such changes would be limited if they are surrounded by stabilization features. Indeed, despite this transition in Delta-Notch signaling, we find that the GRNs in both taxa converge to a conserved regulatory subcircuit that directs the fate of coelomic mesoderm. Therefore, in contrast to previous expectations, evolutionarily stable network kernels that were proposed to function to lock down early developmental regulatory states[2], are not restricted according to network hierarchy. We propose instead that these network subcircuits act as stabilizing features throughout development, functioning as developmental checkpoints through which embryogenesis must transit (Fig. 6B).

## Methods

**Animal and embryo handling**. Adult *Patiria miniata* were obtained from the southern coast of California, USA (Pete Halmay or Marinus Scientific) and were used to initiate embryo cultures following the protocol by Cheatle Jarvela and Hinman[48]. Briefly, testes and ovaries dissected from adult *P. miniata* and oocytes and sperm were isolated. Oocytes were allowed to mature in artificial seawater (ASW) plus 10 μM 1-methyladenine (Spectrum Chemical, product # M3096) for 45 min prior to fertilization. Embryos were cultured in ASW at 16 °C with occasional mixing by agitation.

**Whole-mount staining**. Embryos were fixed and in situ hybridization was performed following the protocol of Hinman, Nguyen and Davidson[49]. Briefly, embryos were fixed in a solution of 4% paraformaldehyde in MOPS-fix buffer (0.1 M MOPS pH 7.5, 2 mM MgSO4, 1 mM EGTA, and 800 mM NaCl) for 90 minutes at 25 °C and transferred to a solution of 70% ethanol for long term storage at −20 °C. In situ hybridization experiments were performed using digoxigenin-labeled antisense RNA probes hybridized at 60 °C. Probes were designed using gene model sequence predictions from legacy.echinobase.org[50,51]. Detection was performed using an anti-digoxigenin AP-conjugate antibody (Roche Cat# 11093274910) followed by an NBT/BCIP reaction (Roche). For two-color FISH, a second dinitrophenyl-labeled antisense RNA probe was hybridized simultaneously[35] and probes were detected using both anti-digoxigenin POD-conjugate antibody (Roche Cat# 11207733910, RRID:AB_514500) and anti-DNP HRP-conjugate antibody (Perkin Elmer Cat# FP1129) followed by tyramide signal amplification (Perkin Elmer). Images of colorimetric whole-mount specimens were taken using a Leica DMI 4000B microscope equipped with a Leica DFC 420C camera and fluorescent specimens were photographed using a Zeiss LSM 880 scanning laser confocal microscope. At least two independent biological replicate experiments were performed for each in situ staining experiment, examining the pattern of at least 10 specimens per replicate.

**Quantitative PCR**. RNA was extracted using the GenElute Mammalian Total RNA Kit (Sigma-Aldrich) and DNA was removed using the DNA-free™ DNA Removal Kit (Invitrogen). Quantitative real-time PCR was performed using the qScript One-Step SYBR Green qRT-PCR Kit (QuantaBio) and the and run on an Applied Biosystems 7300 Real-Time PCR instrument. The sequence of all qPCR primers used is reported in Supplementary Table 2. Measured Ct values for reported genes were normalized to the Ct of an internal control *lamin2β receptor* (GenBank ID: KJ814251.1[52]).

**Perturbation of gene expression**. Zygotes were injected with morpholino antisense oligonucleotides (MASOs; GeneTools) following the protocol by Cheatle Jarvela and Hinman[48]. For all MASOs, the GeneTools standard control MASO was injected into sibling embryos. The observed phenotype of each MASO knockdown was confirmed by injecting a second MASO designed to the same transcript. The sequence and effective concentration used for each MASO used is reported in Supplementary Table 1. Notch perturbations were achieved by bathing embryos in 32 μM DAPT[28] or dimethyl sulfoxide as a control from the two-cell stage. WMISH was performed on at least three independent sets of perturbed embryos. At least ten embryos were assessed in each replicate and phenotypes were counted and a summary is reported in Supplementary Table 3. Quantitative measures of perturbation were achieved by performing qPCR on perturbed compared with control siblings. Each assay was performed on at least two qPCR replicates in each of two biological replicates. Log2-transformed fold-change values between control and experimental groups are shown.

**Gene regulatory network construction**. The GRN model depicting sea star endomesoderm was constructed using BioTapestry[53,54]. The network was constructed by reviewing literature, spanning the years 2003–2019, which describes both embryonic gene expression and gene regulation in *Patiria miniata*. The experimental evidence supporting each node and edge is provided in Supplementary Data 1 and all references to work cited in the GRN experimental evidence utilized are herein cited[19,35,38,41,45,47,52,55–63]. The expression and regulatory linkages are included as reported and have been ordered according to the embryonic chronology and spatially arranged into appropriate territories. A summary of these findings is presented in Supplementary Fig. 1, and a dynamic and interactive model of a more detailed network is hosted on the web at http://grns.BioTapestry.org/PmEndomes[64] for further and more fine-grained exploration of the GRN[64]. Additionally, a BioTapestry.btp file is included in the supplementary materials (Supplementary Data 2) and can be viewed using the BioTapestry desktop application available for download at http://www.BioTapestry.org. A brief guide of how to navigate the BioTapestry user interface is provided (Supplementary Fig. 2). We expect that the online model will continue to be updated in the future to capture changes to the network.

There are three principal temporal subdivisions that span 10–24 hpf, 25–34 hpf, and 35–50 hpf, the breakpoints relating to major embryonic milestones, i.e., the distinction of mesoderm from endoderm at ~24 hpf and subsequently the split between mesenchymal and coelomic-fated mesoderm at ~35 hpf. The BioTapestry user can select which of these models to view by clicking on it in the left-hand panel of the viewer. The models are organized in a hierarchy, with the top-level Full Genome model showing all nodes and links present in all the submodels. The three submodels, representing the three principal temporal subdivisions listed above, summarize the behavior of the network in the various modeled developmental domains that exist for that time period. Below each of these models in the hierarchy are dynamic models that show hourly views (using the time slider in the lower left) for that period. Note that though the time slider is hourly, expression states between the experimental data points (5 h apart) are being interpolated. The placement of data points at five-hour intervals was based on the availability of transcriptomic data spaced roughly at these intervals[19] and previous studies surveying early endomesodermal specification during five-hour intervals[47].

The BioTapestry model is designed to summarize the known information about the *P. miniata* developmental GRN, obtained both from literature and from our experimental results, and it is crucial to use the interactive online version to best understand the behavior of the network. The differential temporal and spatial expression patterns of the genes in the network, as determined by experiment and known with a high degree of confidence, are depicted by showing the genes as "on" or "off" (colored or gray, respectively) in the various regions of the model at each timepoint. In nodes where variable levels of expression are crucial (viz. the gradient of nuclearized beta catenin/TCF from Mesoderm to Veg1 Ectoderm that is present in the early Endomesoderm 10–24 h summary model) the nodes are depicted using intermediate levels of gray to colored. Note that by right-clicking on a gene name and selecting Experimental Data from the pop-up menu, you can view the underlying experimental expression data for the gene, as well as experimental data supporting inputs to the gene. The edges of the network are also based upon literature and experimental results. Of course, there are many different levels of confidence that can be assigned to each edge, based upon the type of experiment, and colored diamonds below the link terminus on target genes are used to indicate confidence. The highest confidence based upon detailed cis-regulatory analysis of a target gene[61], is depicted with a green diamond, see, e.g., Tbr activation of *otx* in the GRN model. However, most links in the network are backed by the results of perturbation experiments (e.g., MASO knockdown of the source gene or drug perturbation of a signaling pathway). To ensure only high confidence links are included, we use a threshold of at least twofold change observed in a minimum of two independent perturbation experiments. We also utilize multiple MASOs targeting the same transcript to ensure specificity of the observed phenotypes (see Supplementary Table 1). While there is no guarantee that these links are in fact direct, direct edges that can be explained through an indirect path can be omitted through a parsimonious approach to adding links to the network.

Links, like nodes, are also shown as "on" or "off" in the model at each point in space and time, simply based on the expression of the source gene at that same point. Notably, this depiction says nothing explicit about the actual cis-regulatory logic that is encoded in the target gene. Just because a link is shown as colored and incident on a target gene does not mean that it has been shown to be necessary at that point in space and time to cause expression of the target gene. To make that conclusion, much more targeted experiments are required to make that claim. However, the on/off state of the target gene and the inbound links can provide clues to generate hypotheses and suggest further experiments. For example, if all the links into a gray (off) target are colored (on), that suggests that there must be other unknown inputs into the target gene.

This model is not purporting to be complete but is instead a systems-level summary of the existing state of knowledge about the causal mechanisms underpinning *P. miniata* development driven by the GRN. It is certainly missing genes, and in fact since it is heavily based on orthology to genes present within the sea urchin GRN, we expect this network is biased towards including just those transcription factors. Furthermore, it has not been validated by computational simulations, and involves no detailed modeling of the transcriptional mechanisms that control gene expression. In this regard, it is like the sea urchin network, which was first developed using gene expression and perturbation data[9] many years before boolean simulations were performed to ascertain the ability of the model to explain the observed behavior[11].

**Reporting summary**. Further information on research design is available in the Nature Research Reporting Summary linked to this article.

## Data availability
The authors declare that all data supporting the findings of this study are available within the article and its supplementary information files or from the corresponding author upon reasonable request. The data used to synthesize the network models presented in this paper are summarized in an online resource hosted at http://grns.BioTapestry.org/PmEndomes/. The interactive network visualization accessible through this URL provides detail about all the experimental evidence supporting the expression timing and localization of each node in the network as well as experimental perturbations to support network edges. Each piece of data is cited to original publications for further assessment.

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

## Acknowledgements

We thank Dr. William Hatleberg for helpful feedback during the preparation of the manuscript. This work was supported by the Binational Science Foundation grant number 2015031 to V.F.H., the National Science Foundation grants IOS 1557431 and MCB 1715721 to V.F.H. and the National Institute of Health grant P41HD071837 to V.F.H.

## Author contributions

G.A.C., B.S.M., and V.F.H. conceived of and designed experiments. G.A.C., B.S.M., O.Z. and J.P. carried out experiments. G.A.C. and B.S.M. analyzed data and V.F.H. was instrumental in the interpretation of the results. G.A.C. and W.L. constructed the network model. G.A.C. wrote the manuscript with significant input from B.S.M. and V.F.H. All authors read and approved of the final manuscript.

## Competing interests

The authors declare no competing interests.
