## [Peer Review File · Nature Communications]

Reviewers' Comments:

Reviewer #1:

Remarks to the Author:

This paper interrogates the gene regulatory (GRN) of the seastar, *Patiria miniata*, in a study that is parallel to that performed in sea urchin from Eric Davidson's lab and published in 2008. The idea is to test between the two species whether specific gene interactions are similar or different. For this purpose they use in situ and perturbation experiments to test gene interaction.

In my original review of this paper, I asked for better controls. The authors rightly argue that they wanted to draw parallels between the sea star and sea urchin and therefore used the same approaches and essentially repeat all of the urchin experiments in a different species. They find that using two separate morpholinos per knockdown produces the same phenotype. While this may be standard in the sea urchin community as a control, I still have a problem with using decades old technology given the problems with morpholinos in many organisms (e.g. zebrafish). It would be much better to use CRISPR which seems to work well in all organisms, at least as a secondary knock-down method. Similarly, it would be nice to have a secondary validation to the DAPT inhibitor experiments. Other than that, the authors have addressed my major concerns and the paper seems appropriate for publication in *Nature Communications*, though it might profit from being rewritten in a longer format.

Reviewer #2:

Remarks to the Author:

The authors have addressed my concerns to my satisfaction. The current version of this ms is still highly technical but more accessible and appropriate for the readership of *Nature Communications*.

Two very minor issues I noted:

line 174: lockdown -> lock down

line 375: "edges are regulated..." Rephrase. An edge is not regulated, because it is just a symbol indicating interactions between two molecules or genes.

Reviewer #3:

Remarks to the Author:

I have reviewed the previous version of this manuscript and have carefully read the new version of it and the response of the authors to the the four reviewers' concerns.

I am very satisfied with this new version and with the explanation given by the authors who, in my opinion, addressed all my and other reviewers' concerns. In particular, the addition of the the new supplemental table (Extended Data, Table 4), the update of the BioTapestry on line tool and the addition of new experiments to confirm some important nodes of the sea star endomesodermal network. Also the reading of the manuscript, simplified of too many jargon terms and with the

addition of some introduction/ explanation to echinoderm development is much smoother and better suited for a general audience. On the other end, the on line tool provided at the BioTapestry web site allows the reader who wish to dive in all the details of the analysis to fully appreciate the work beyond the GRN presented and discussed in the main manuscript.

Two general points stays:

1) The major point is the fact that perturbation experiments without cis-regulatory analysis give no insight into whether the interactions are direct or indirect, and this makes more difficult to draw comparable topologies, especially in cases where the sea urchin has cis-regulatory analysis done. However, this said, I agree with the authors on the fact that this GRN, as any other "complete" GRN, can be improved over time and yet the conclusions the authors reached at this time are very solid and represent a significant contribution to the understanding of evolution of novelties

2) On the actual BioTapestry page, the time can be selected by hour, but the actual experimental data is only for 5 hour windows so what is shown is interpolated. It is not clear to me if this was based on any time-course transcriptome, but in any case, the choice made should be more explicit.

Reviewer #4:

Remarks to the Author:

The authors have extensively revised their original manuscript, adding new experimental evidence, clarifying criteria for inferring interactions, defining important terms for the non-specialist, and removing some confusing and contentious statements. The manuscript is now clearer and more accessible to readers outside the sea urchin systems biology community. I remain a bit underwhelmed by the evolutionary conclusions, which still seem rather vague. That being said, there is now an interesting proposal presented that conserved network kernels act to stabilize evolutionary changes in signaling modes. It's not entirely clear to me how such a proposal could be tested rigorously, but hopefully the authors or someone else will think of a way as this would represent an important step towards understanding how developmental GRNs evolve. Overall, the authors have made a good faith effort to meet the concerns that I raised and I do not see any need to further delay publication of this study.

Reviewer #1 (Remarks to the Author):

This paper interrogates the gene regulatory (GRN) of the seastar, *Patiria miniata*, in a study that is parallel to that performed in sea urchin from Eric Davidson's lab and published in 2008. The idea is to test between the two species whether specific gene interactions are similar or different. For this purpose they use in situ and perturbation experiments to test gene interaction.

In my original review of this paper, I asked for better controls. The authors rightly argue that they wanted to draw parallels between the sea star and sea urchin and therefore used the same approaches and essentially repeat all of the urchin experiments in a different species. They find that using two separate morpholinos per knockdown produces the same phenotype. While this may be standard in the sea urchin community as a control, I still have a problem with using decades old technology given the problems with morpholinos in many organisms (e.g. zebrafish). It would be much better to use CRISPR which seems to work well in all organisms, at least as a secondary knock-down method. Similarly, it would be nice to have a secondary validation to the DAPT inhibitor experiments. Other than that, the authors have addressed my major concerns and the paper seems appropriate for publication in Nature Communications, though it might profit from begin rewritten in a longer format.

We agree and thank the reviewer for this comment. While it is beyond the scope of the current work to include these alternative perturbation strategies in the future. To further highlight this as a limitation we have added the following sentence to the manuscript:

Although gene perturbation by morpholino knockdown has historically been a useful tool for disentangling these networks, it is important to recognize that the links drawn are largely provisional until fully borne out by independent perturbation methods (e.g. CRISPR) and cis regulatory analyses to test whether the indicated interactions are indeed direct.

Reviewer #2 (Remarks to the Author):

The authors have addressed my concerns to my satisfaction. The current version of this ms is still highly technical but more accessible and appropriate for the readership of Nature Communications.

Two very minor issues I noted:

line 174: lockdown -> lock down

line 375: "edges are regulated..." Rephrase. An edge is not regulated, because it is just a symbol indicating interactions between two molecules or genes.

We have corrected both of these issues. Thank you for bringing them to our attention.

Reviewer #3 (Remarks to the Author):

I have reviewed the previous version of this manuscript and have carefully read the new version of it and the response of the authors to the the four reviewers' concerns.

I am very satisfied with this new version and with the explanation given by the authors who, in my opinion, addressed all my and other reviewers' concerns. In particular, the addition of the the new supplemental table (Extended Data, Table 4), the update of the BioTapestry on line tool and the addition of new experiments to confirm some important nodes of the sea star endomesodermal network. Also the reading of the manuscript, simplified of too many jargon terms and with the addition of some introduction/ explanation to echinoderm development is much smoother and better suited for a general audience. On the other end, the on line tool provided at the BioTapestry web site allows the reader who wish to dive in all the details of the analysis to fully appreciate the work beyond the GRN presented and discussed in the main manuscript.

Two general points stays:

1) The major point is the fact that perturbation experiments without cis-regulatory analysis give no insight into whether the interactions are direct or indirect, and this makes more difficult to draw comparable topologies, especially in cases where the sea urchin has cis-regulatory analysis done. However, this said, I agree with the authors on the fact that this GRN, as any other "complete" GRN, can be improved over time and yet the conclusions the authors reached at this time are very solid and represent a significant contribution to the understanding of evolution of novelties

We agree and thank Dr Arnone for this comment. While it is beyond the scope of the current work to include extensive cis-regulatory analyses of these edges, we have added the following sentence to the manuscript to highlight the need for additional experiments to root out which of these interactions are truly direct:

Although gene perturbation by morpholino knockdown has been a useful tool for disentangling these networks, it is important to recognize that the links drawn are largely provisional until fully borne out by independent perturbation methods (e.g. CRISPR) and cis regulatory analyses to test whether the indicated interactions are indeed direct.

2) On the actual BioTapestry page, the time can be selected by hour, but the actual experimental data is only for 5 hour windows so what is shown is interpolated. It is not clear to me if this was based on any time-course transcriptome, but in any case, the choice made should be more explicit.

Thanks for the comment and have added the following to the materials and methods to clarify the choice made:

Note that though the time slider is hourly, expression states between the experimental data points (five hours apart) are being interpolated. The placement of data points at five-hour intervals was based on the availability of transcriptomic data spaced roughly at these intervals¹⁹ and previous studies surveying early endomesodermal specification during five-hour intervals⁴⁷.

Maria ina Arnone

Reviewer #4 (Remarks to the Author):

The authors have extensively revised their original manuscript, adding new experimental evidence, clarifying criteria for inferring interactions, defining important terms for the non-specialist, and removing some confusing and contentious statements. The manuscript is now clearer and more accessible to readers outside the sea urchin systems biology community. I remain a bit underwhelmed by the evolutionary conclusions, which still seem rather vague. That being said, there is now an interesting proposal presented that conserved network kernels act to stabilize evolutionary changes in signaling modes. It's not entirely clear to me how such a proposal could be tested rigorously, but hopefully the authors or someone else will think of a way as this would represent an important step towards understanding how developmental GRNs evolve. Overall, the authors have made a good faith effort to meet the concerns that I raised and I do not see any need to further delay publication of this study.

We thank the reviewer for their thorough and thoughtful commentary on this work.